# Pension levels of chinese institutions during the transition period: A case study of universities

**Xu-Juan Lu[1]\***, **Qian-Xing Zhuang**[ORCID][2]\*

**1** Department of Human Resources, Nantong University, Nantong, Jiangsu, China, **2** Department of Physiology, School of Medicine, Nantong University, Nantong, Jiangsu, China

\* lxj625218@ntu.edu.cn (XJL) qxzhuang@ntu.edu.cn (QXZ)

## Abstract

### Background

With China's rapidly aging population, the combined effects of this demographic shift and a growing wave of retirements present significant challenges for the sustainability of the basic pension system. To address these issues, China initiated reforms to the basic pension insurance system for institutions, including universities, on October 1, 2014, establishing a ten-year transition period. As this period nears its end, it is essential to analyze changes in retirement benefits for institutional retirees over the past decade.

### Materials and methods

This study used representative data from retired employees in various post categories at Jiangsu universities, covering two periods: before the transition (prior to September 30, 2014) and during the transition (October 1, 2014, to December 31, 2023). The dataset included 1,200 retirees before the transition and 3,720 retirees during the transition.

### Results

The income of retired university employees rose significantly before and during the transition period, with transition incomes surpassing those from earlier phases and stabilizing afterward. This increase was mainly due to the introduction of the occupational annuity system. Factors such as gender, retirement year, retirement age, payment index, post category, and level affected pension payment levels to varying extents. Notably, cumulative years of payment influenced by gender and other factors had the greatest impact on these levels.

### Conclusions

The implementation of the occupational annuity system during the transition period revealed certain shortcomings in the endowment insurance reform. The transitional

**Data availability statement:** The data supporting the findings reported in this study were obtained from the Jiangsu Provincial Department of Human Resources and Social Security (DHRSS). Given that this study involves sensitive information related to personal income, a high level of confidentiality is imperative. As employees engaged in social security operations at Nantong University (No.: 1032833767), we have the access to utilize such data; however, we are obligated to adhere to stringent confidentiality standards and therefore do not have the authority to share these data. Qualified academic researchers may request access to a minimal dataset solely for the purpose of reproducing this study, provided that such data transfer complies with DHRSS regulations regarding general data sharing. For further information, please contact administrator of the Jiangsu smart people society at Nantong University, Email: lzk@ntu.edu.cn.

**Funding:** This work was supported by the General Projects of Philosophy and Social Sciences in Higher Education Institutions of Jiangsu Province (grants 2025SJYB1255). The funders had no role in study design, data collection and analysis, decision to publish, or preparation of the manuscript.

**Competing interests:** The authors have declared that no competing interests exist.

pension from the Social Collective Fund represents a significant portion of total pension expenditure, creating a heavy financial burden on the fund. Additionally, extending and standardizing retirement age for men and women can help reduce gender income disparities. Delayed retirement alleviates pressure on social pension systems by increasing individual incomes and aligning pension levels with societal needs. This approach promotes social equity and provides adequate care for older adults.

## 1. Introduction

The sustainability of old age insurance systems remains a critical challenge for both developed and developing nations [1,2]. To establish a more equitable, sustainable, and unified social pension insurance system, the Chinese government initiated a pension insurance system for public institutions on October 1, 2014. These reforms are guided by the principles of comprehensive coverage, basic insurance provision, multitiered development, and sustainability, aiming to enhance equity, address liquidity needs, and ensure long-term viability. A key objective is to align the pension insurance model for institutions with that for enterprises, thereby eliminating disparities in pension calculations between retirees from enterprises and institutions [3]. Under this new policy, public institutions have now adopted a basic pension insurance system consistent with that of enterprise workers, combining an institutional planning model with individual accounts. Payments are shared between institutions and individuals at specified ratios. Moreover, reforms in pension planning and distribution methods have been introduced along with a basic pension adjustment mechanism and the implementation of an occupational annuity system.

Since the implementation of the pension reform, individuals who commenced employment before September 30, 2014 and retired after October 1, 2014 have entered a transitional phase between the old and new systems. To protect the rights of retirees from public institutions, a 10-year transitional period (October 1, 2014, to September 30, 2024) was established, during which a unified retirement approach was adopted. The retirement benefits during this period are governed by the "high guaranteed bottom limit" principle: if an individual's planned benefits (including occupational annuity provisions) fall below the pre-transition standards, payments can be adjusted to meet those standards; if the benefits exceed the pre-transition standards, a ceiling can be applied. As the 10-year transition period approaches its conclusion, it becomes critical to assess changes in retiree benefits before and during this phase, identify the factors driving these changes, and determine the most influential elements. In addition, the impact of the occupational annuity system on retiree benefits during this period warrants further examination.

Chinese universities are primarily public institutions. This study employed empirical analysis of survey data from retired employees at these institutions to explore the issues outlined above. A random sample of 1,200 retirees from Jiangsu Province who retired before September 30, 2014 (pre-transition period) and 3,270 retirees who retired between October 1, 2014, and December 31, 2023 (transition period)

was selected. First, the analysis compared the total pension income of employees across the two periods. The findings revealed a significant increase in total pension income during both periods, with the difference between them remaining relatively stable. The occupational annuity policy implemented during the transition period has been identified as a key factor contributing to disparities in pension outcomes. To further explore these differences, statistical analyses were conducted to examine the factors influencing basic pension calculations during the transition period, including gender, retirement year, post category, and grade. The findings revealed that gender significantly affected payment years, actual average payment index, personal account accumulations, and occupational annuity balances among male and female employees within the same retirement year. Additionally, variations in retirement years between male and female employees of the same age substantially affected the overall pension income. The disparities in post categories and grades also contributed to the imbalances in planned payment months, actual average payment index, basic pensions, personal account pensions, transitional pensions, and occupational annuities, further influencing retirees' overall pension income during the transition period. These findings could provide valuable insights for Chinese social institutions to develop and refine their retirement systems beyond the transition period.

## 2. Materials and methods

### 2.1. Samples

University personnel were classified into three categories: ground-skilled staff, management staff, and professional and technical staff. This study randomly selected a sample of 4,470 retired employees from higher education institutions in Jiangsu Province, using data obtained from the Department of Human Resources and Social Security on August 9, 2024. Among them, 1200 employees retired before September 30, 2014, comprising 200 ground-skilled staff, 400 management personnel, and 600 professional and technical employees. During the transition period (October 1, 2014, to December 31, 2023), 3,720 employees retired, consisting of 620 ground-skilled staff (from grades I to IV), 1,240 management personnel (from grades IV to IX), and 1,860 professional and technical employees (from grades II to X).

### 2.2. Pension measurement approaches prior to and during the transition period

**2.2.1. Total pension prior to the transition period.** Retirement benefits for staff members of public institutions were calculated using the following formula:

$$TP_{PT} = S_R P_D + S_{SR}$$

where $TP_{PT}$ denotes the total pension prior to the transition period, and $S_R$ signifies the salary at the time of retirement based on their pre-retirement salary level and years of service. $P_D$ represents the proportion of distribution: employees with 30–35 years received 85%, and those with over 20 but less than 30 years of service were entitled to 80%. All salary standards adhered to 2006 salary regulations for Chinese public institutions. $S_{SR}$ denotes the subsidy standard for retirement, which indicates an increase in retirement expenses in accordance with the state regulations.

**Total pension during the transition period.** The total pension during the transition period was calculated using the following formula:

$$TP_{DT} = B_P + OA$$

where $TP_{DT}$ denotes the total pension during the transition period, and $B_P$ represents the basic pension, with actual payments adjusted according to the year of retirement. OA denotes occupational annuity, which includes payments made by institutions and individuals, investment and operation of occupational annuity funds, and other income as stipulated by the state. Among them, $B_P$ was calculated using the following formula:

$$B_P = S_{BP} + T_P + PA_P$$

where $S_{BP}$, TP, and PAP denote the standard basic pension, transitional pension, and personal account pension, respectively. The calculations for $S_{BP}$, $T_P$, and $PA_P$ were further defined using the following formulas:

$$SBP = A_{MS} \frac{1 + E_{PI} \, E_{PY} + A_{PI} \, A_{PY}}{2} (E_{PY} + A_{PY}) \, 1\%$$

$$API = \frac{\left( \frac{Xn}{Cn-1} + \frac{Xn-1}{Cn-2} + \cdots\cdots + \frac{X2015}{C2014} + \frac{X2014}{C2013} \right)}{A_{PY}}$$

$$T_P = A_{MS} \, E_{PI} \, E_{PY} \, TC$$

$$PAP = \frac{T_{PA}}{P_{PM}}$$

where $A_{MS}$ denotes the average monthly salary of local employees in the previous year at the time of retirement, $E_{PI}$ represents the equivalent payment index, $E_{PY}$ represents the equivalent payment years, $A_{PY}$ denotes the actual payment years, and $A_{PI}$ denotes the actual average payment index. Among them, $X_n$, $X_{n-1}$,..., $X_{2014}$ represent the cumulative salary base of personnel from their retirement year back to 2014, and $C_{n-1}$, $C_{n-2}$,..., $C_{2013}$ denote the annual average salary of local employees for each corresponding year from one year prior to retirement back 2013.

TC denotes the transition coefficient, which is consistent with that applied to basic pension insurance for enterprise employees in pension insurance pooling areas designated for institutions, and is formulated by each respective region. $T_{PA}$ denotes the total accumulated amount in the personal account of basic pension insurance at the time of retirement. $P_{PM}$ represents the planned payment months, with the duration of these months determined in accordance with national regulations and calculated from the month following retirement until an individual reaches China's average life expectancy.

### 2.3. Statistical analysis

All statistical analyses were performed using SPSS version 17.0, with the data presented as the mean ± standard deviation (SD). A two-tailed unpaired t-test was used for comparisons between two groups, while one-way analysis of variance (ANOVA) followed by Student-Newman-Keuls (SNK) post hoc testing was used for comparisons involving three or more groups. Pearson's correlation test was used to evaluate the relationships between the datasets. The significance levels were indicated as *$P < 0.05$, **$P < 0.01$, and ***$P < 0.001$, with $P < 0.05$ representing a statistically significant difference and $P > 0.05$ indicating no significant difference (denoted as ns). The comparison of objects, the statistical methods employed, and the group size ($n$), which indicates independent values, are all presented in (S1 Table).

### 3. Results

### 3.1. Variations in the overall pension of university retirees before and during the transition period

The experiment was conducted by comparing the overall pension per capita and the differences between pre-transition and transition retirees in 2014 (the onset of the transition period) and 2023 (approaching its conclusion). The findings indicated that in 2023, the overall pension per capita for both groups of retirees was higher than that in 2014 (Fig 1A).

Furthermore, the disparity between the two groups remained relatively stable over this period (Fig 1B). A year-on-year analysis of the overall pension per capita from 2014 to 2023 revealed that transition retirees consistently received higher pensions than their pre-transition counterparts did throughout the entire transition period (Fig 1B). Notably, the difference between the two groups demonstrated little variation from 2014 to 2023, maintaining an approximate level of 2520 RMB/month during the transition period (Fig 1B).

The overall pension for retirees during the transition period included both a basic pension and occupational annuity, whereas the overall pension for pre-transition retirees did not include an occupational annuity. To examine this further, we analyzed the trends in basic pensions and occupational annuities for transition-period retirees and compared them with the per capita overall pensions of pre-transition retirees. The results indicated that both the average basic pension and occupational annuities for transitional retirees increased annually (Fig 1C). Notably, the basic pension for transition retirees consistently surpassed the overall pension of pre-transition retirees, while the gap gradually narrowed. At the current rate, parity is projected to be achieved by 2035 (Fig 1C).

In 2014, the average overall pension for pre-transition retirees was 4707.57±429.41 RMB/month, while the average basic pension for transition-period retirees was 6461.93±1661.90 RMB/month, resulting in a gap of 1754.36 RMB/month between the two groups. By 2023, the average overall pension for pre-transition retirees increased to 9535.33±1660.82 RMB/month, and the basic pension for transition-period retirees rose to 10498.49±2193.95 RMB/month, reducing the gap to 963.16 RMB/month. Since October 1, 2014, occupational annuity policies have been applied only to transition-period retirees. As occupational annuities increased annually, the anticipated reduction or elimination of pension disparities

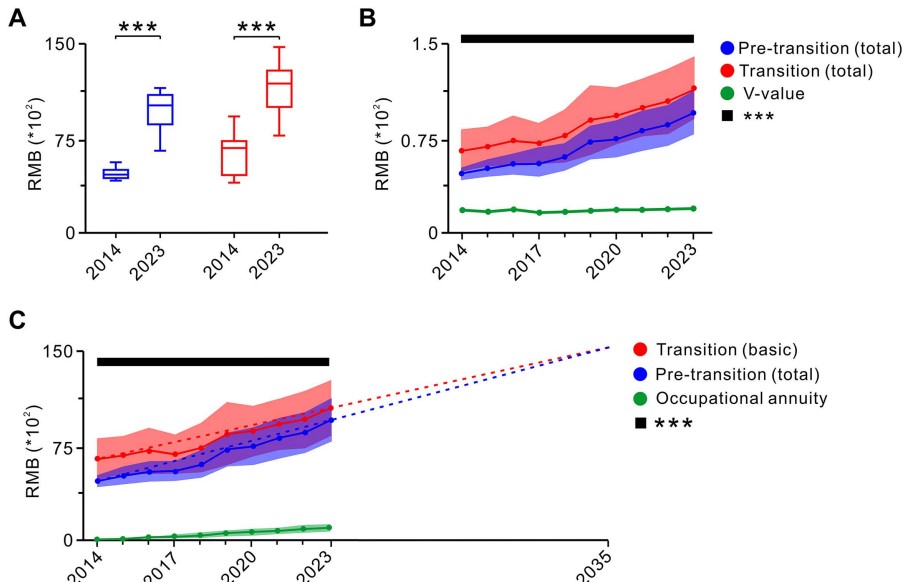

**Fig 1. Overall pension of retirees before and during the transition period from 2014 to 2023 as well as the changes in basic pensions and occupational annuities for retirees throughout this period. A:** A comparison of total pension benefits for retirees prior to and during the transition period, highlighting the differences between these two periods in both the beginning year of the transition (2014) and the concluding year (2023) (two-tailed unpaired *t*-test). **B:** Analysis of total pension benefits comparing pre-transition retirees with transition retirees along with an examination of how this difference evolves on a yearly basis (two-tailed unpaired *t*-test). **C:** Annual variations in basic pensions for retirees during the transition period compared with the overall pensions received by retirees before this period, as well as occupational annuities during the transition period (two-tailed unpaired *t*-test). Pre-transition (total), total pension benefits for retirees prior to the transition period; Transition (total), total pension benefits for retirees during the transition period; V-value, variations in total pension benefits for retirees prior to and during the transition period; Transition (basic), basic pension benefits for retirees prior to the transition period. Data are presented as mean±standard deviation; ns indicates no significant difference; ***$P<0.001$.

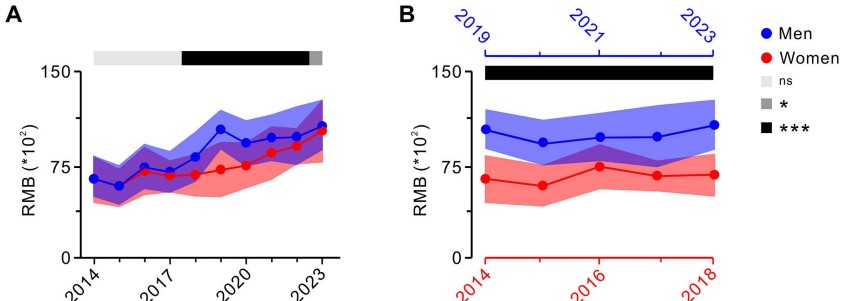

between pre-transition and transition retirees has not been observed. Instead, throughout the transition period, a significant and relatively stable gap in per capita overall pensions persisted between the two groups.

### 3.2. Impact of gender and retirement year on the basic pension of university retirees during the transition period

The impact of pension policies on retirees' pensions during the transition period is a critical issue for numerous in-service personnel as it can directly affect their post-retirement quality of life. Additionally, this issue provides valuable insights for governments to formulate and adjust pension policies beyond the transition period. This study examined how gender and retirement age affect basic pensions during the transition phase. Under China's current retirement policy, on average, men retire five years later than women. As a result, men typically retire later than women of the same age group, which is a key factor contributing to disparities in basic pension amounts.

When men and women retire in the same year, the calculation formula for the basic pension during the transition period indicates that the positive factors influencing men's pensions, such as the average monthly salary of local employees in the previous year, actual payment years, actual average payment index, accumulated personal account balances, and occupational pension accumulations at retirement, are generally more favorable than those for women. Conversely, men face a disadvantage compared to women in terms of negative factors such as planned payment months. These differences can collectively result in men receiving a higher average basic pension than women retiring in the same year (Fig 2A).

When men and women of the same age reach their statutory retirement age at different times, disparities in their basic pension become more pronounced. For example, a woman who reached retirement age in 2014 would have a male counterpart who did not retire until 2019. During the transition period, the calculation formula for the basic pension indicates that the positive and negative factors influencing the pension remain consistent when male and female employees retire simultaneously. According to data from China's National Bureau of Statistics, the average monthly salary in Jiangsu Province rose from 5,149 yuan a decade ago to 8,623 yuan a year. Consequently, males who retire later benefit from higher average monthly salaries, resulting in an even greater average basic pension for men than for women when they retire at their respective statutory ages (Fig 2B).

Promoting a voluntary and flexible retirement policy along with the gradual and orderly implementation of a delayed legal retirement age and advocating for equal retirement ages for men and women can effectively address inconsistencies in retirement years caused by gender differences. This approach would help reduce disparities in basic pension payments and achieve equity in basic pension income between men and women after retirement.

**Fig 2. Annual variations in basic pensions for retired men and women during the transition period. A:** Disparity in the basic pension between men and women who retire in the same year (two-tailed unpaired $t$-test). **B:** Difference in basic pensions between men and women of identical age within the same year upon reaching their respective statutory retirement age (two-tailed unpaired $t$-test). Data are expressed as mean±SD; ns indicates no significant difference; $*P < 0.05$, $***P < 0.001$.

### 3.3. Impact of gender and retirement year on occupational annuities for university retirees during transition

The overall pension during the transition period comprises a basic pension and an occupational annuity. According to China's retirement regulations, the occupational annuity fund is derived from payments made by universities and individuals, income generated from the fund's investment and operations, and other income stipulated by the state. Moreover, the factors influencing the payment base of occupational annuity are closely linked to the gender and retirement year of retirees after the reform implementation date (September 30, 2014). Building on the analysis of the impact of gender and retirement year on basic pension, this study further examined their effects on occupational annuity.

We first analyzed the impact of gender and retirement year on retirees' occupational annuities. The findings revealed that when men and women retired in the same year, men may typically retire later than women. Consequently, there were fewer months in which men could receive occupational annuity payments after retirement than for women. However, this ultimately resulted in a higher occupational annuity income for men than for women (Fig 3A).

When men and women of the same age reached their respective statutory retirement ages, the factors affecting the number of months of occupational annuity payments remained consistent with those observed when they retired in the same year. However, during the transition period, men had significantly more years of occupational annuity payments than women. As a result, men's occupational annuity income was even higher than that of women when they reached their statutory retirement age separately (Fig 3B). This income disparity reflected the principle of "paying more and getting more, paying more and getting more for longer" embedded in the reform of the pension insurance system. Therefore, postponing retirement could effectively enhance individual occupational annuity income, whereas simultaneous retirement for men and women in the same cohort could help eliminate gender-based disparities in occupational annuity income.

### 3.4. Post category and level on the basic pension of university retirees during the transition period

We analyzed the impact of post category and level on basic pensions (Table 1) and discovered a significant relationship between monthly basic pension amounts and both post category and level. The statistical data demonstrated that the ground-skilled staff received the lowest average basic pension at $5548.98 \pm 1268.24$ RMB, followed by the management staff at $7520.50 \pm 1781.93$ RMB, while the professional and technical staff had the highest average at $9847.67 \pm 2146.93$ RMB. Furthermore, within each post category, the average basic pension increased with higher post levels. According to the basic pension calculation formula during the transition period, factors such as planned payment months, actual average payment index, base pensions, personal account pensions, and transitional pensions influenced monthly pension payments. We further examined how these factors could collectively affect the overall basic pension.

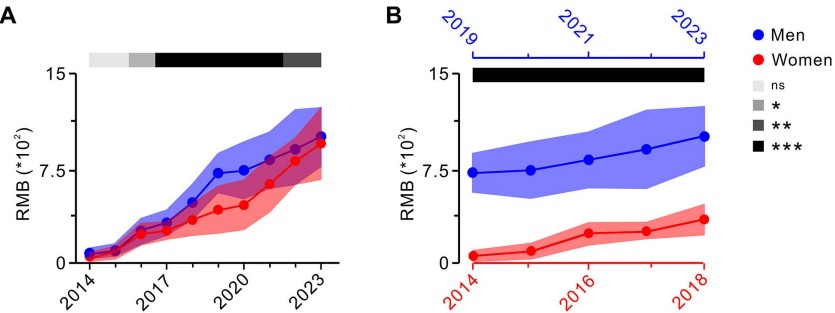

**Fig 3. Annual changes in occupational annuities for retired men and women during the transition period. A:** Difference in basic pensions between men and women who retired in the same year (two-tailed unpaired *t*-test). **B:** Difference in the basic pension of men and women of identical age at the time they reach their statutory retirement age (two-tailed unpaired *t*-test). Data are expressed as mean±SD; ns indicates no significant difference; *P<0.05, **P<0.01, ***P<0.001.

**Table 1. Composition of basic pension for different post categories and post levels during the transition period.**

| Post category | Post level | Planned payment months | Actual average payment index | Standard basic pension | Personal account pension | Transitional pension | Basic pension |
|---|---|---|---|---|---|---|---|
| Professional and technical staff | II | 107.26±14.77 | 2.28±0.10 | 5963.87±677.52 | 1179.35±315.53 | 8443.75±960.70 | 14211.07±1947.64 |
| | III | 121.71±18.98 | 2.15±0.17 | 4680.64±624.35 | 619.5±303.64 | 6837.45±857.43 | 10972.09±1336.37 |
| | IV | 131.62±14.26 | 1.96±0.16 | 4555.12±574.76 | 600.55±303.46 | 6539.06±893.53 | 10652.83±1266.14 |
| | V | 139.25±7.00 | 1.73±0.12 | 4239.93±917.63 | 594.52±255.41 | 5558.52±620.80 | 9687.32±1488.64 |
| | VI | 140.16±5.89 | 1.69±0.13 | 3983.24±615.89 | 565.52±304.31 | 5278.94±650.52 | 8977.86±1488.87 |
| | VII | 145.30±12.51 | 1.68±0.11 | 3886.58±672.34 | 504.94±337.04 | 5048.94±672.16 | 8430.21±1524.78 |
| | VIII | 146.13±13.10 | 1.53±0.12 | 3735.06±581.86 | 396.25±247.55 | 4266.17±497.71 | 8369.24±1429.81 |
| | IX | 149.64±14.79 | 1.41±0.10 | 3439.59±620.85 | 359.18±215.42 | 4411.47±595.27 | 7441.75±1346.00 |
| | X | 161.68±13.82 | 1.31±0.13 | 3292.26±461.67 | 339.43±199.78 | 4102.93±365.36 | 7276.17±1023.80 |
| Management staff | IV | 132.14±1.00 | 2.13±0.17 | 4884.90±620.90 | 638.04±318.77 | 6837.75±594.56 | 10325.83±1034.44 |
| | V | 139.00±0.00 | 1.91±0.17 | 4077.25±533.77 | 552.01±337.17 | 5532.67±464.95 | 8588.87±1204.67 |
| | VI | 150.22±14.96 | 1.57±0.07 | 3989.91±504.73 | 505.68±163.32 | 4967.96±609.54 | 7955.00±983.30 |
| | VII | 151.42±15.22 | 1.48±0.16 | 3595.74±587.61 | 409.86±262.21 | 4163.89±613.15 | 6715.36±1153.49 |
| | VIII | 157.53±15.23 | 1.24±0.12 | 3379.66±585.89 | 307.87±219.99 | 4155.02±395.30 | 5985.93±1213.09 |
| | IX | 157.51±15.31 | 1.14±0.01 | 3302.79±486.40 | 278.96±106.40 | 4027.99±523.84 | 5853.48±985.58 |
| Ground-skilled staff | I | 145.32±15.57 | 1.34±0.13 | 3055.79±694.43 | 253.27±147.80 | 3876.33±853.64 | 6268.45±1059.88 |
| | II | 154.84±22.68 | 1.28±0.11 | 3471.12±650.74 | 328.12±197.84 | 4281.40±609.56 | 6143.05±1279.02 |
| | III | 157.62±24.33 | 1.22±0.10 | 2689.62±682.60 | 180.35±150.43 | 3330.72±739.13 | 5181.44±1119.36 |
| | IV | 195.00±0.00 | 0.95±0.08 | 2262.49±570.50 | 176.25±150.82 | 2442.01±544.11 | 4857.69±1015.30 |

A negative correlation was observed between the number of planned payment months and monthly payment amounts for personal account pensions. As the retirement age increased, the average number of planned payment months decreased. Higher post categories and levels also corresponded to fewer planned payment months. Notably, the professional and technical staff had a higher average retirement age than the management staff, who could retire later than ground-skilled staff. The statistical analysis indicated that the average number of planned payment months was 163.43±26.31 for the ground-skilled staff, 147.39±14.50 for the management staff, and 135.34±18.15 for the professional and technical staff (Table 1). Within each post category, an increase in the post level generally correlated with a decrease in planned payment months.

The actual average payment index was positively correlated with the monthly salary payment base of retired employees, with higher post levels corresponding to a higher salary payment base. Professional and technical staff had the highest salary payment base, followed by management and ground-skilled staff. The average actual average payment index was 1.19±0.18 for the professional and technical staff, 1.61±0.34 for the management staff, and 1.82±0.30 for the ground-skilled staff (Table 1). Furthermore, within each job category, an increase in the post level generally corresponded to a higher payment index.

The standard basic pension was positively correlated with the average monthly salary and payment index of local employees in the year prior to retirement. As noted earlier, the payment index increased with higher post levels. Additionally, differences in post categories can influence the retirement year for individuals of the same age, with those in higher posts typically retiring later. Over the past decade, the average monthly salaries of active workers have consistently increased. Consequently, retirees with higher post-retirement levels and later retirement years generally experience a higher average monthly salary in their final working year. Professional and technical staff tended to have the highest average monthly salary before retirement, followed by management staff, and then ground-skilled staff. Our findings confirmed the following trends. The average standard basic pension for the professional and technical staff was 4296.52±846.41

RMB, for the management staff 3827.94±702.12 RMB, and for the ground-skilled staff 2852.92±785.89 RMB (Table 1). Additionally, within each job category, standard basic pensions significantly increased with higher post levels.

The calculation of personal account pensions is closely related to the accumulated amount in the personal account of the basic pension insurance and the number of planned payment months at retirement. The number of payment months is negatively associated with personal account pensions and decreases with later retirement. The accumulated amount in personal accounts is directly proportional to the payment index and the payment period. For public institution employees, the payment index and periods are significantly influenced by factors such as post category and post level. Our study suggested that the professional and technical staff received the highest average personal account pension at 583.55±338.07 RMB, followed by the management staff at 454.10±294.71 RMB, while the ground-skilled staff received the lowest at 229.97±174.67 RMB (Table 1). Within each post category, higher post levels corresponded to increased personal account pensions. This reflects variations in the cumulative payment periods across post categories. As the payment period increased, the accumulated savings in personal pension insurance accounts increased annually, causing a gradual rise in personal account pensions. This trend aligned with the principle of "contributing more and for longer to receive more". Consequently, delayed retirement extended the actual payment period, leading to higher cumulative savings and increased personal account pensions. These findings demonstrate that implementing a flexible retirement system can enhance individual income levels after retirement.

The transitional pension is positively correlated with the average monthly salary of local employees in the previous year, equivalent payment index, and equivalent payment years. The equivalent payment index is influenced by an individual's post category, post level, and pre-retirement salary and also increases with longer service periods. Thus, individuals in higher positions, with greater ranks, salaries, and tenures, tend to receive higher transitional pensions. Our study indicated that the professional and technical staff received the highest transitional pensions, averaging 5861.92±1340.46 RMB, followed by the management staff at 4882.30±991.08 RMB, and the ground-skilled staff at 3461.20±962.70 RMB (Table 1). Within the same post category, transitional pensions increased with higher post levels. The transitional pensions were financed by the Social Collective Fund, in which a higher equivalent payment index led to a higher transitional pension. However, excessively high transitional pensions exacerbated the shortage of China's social pension funds, further straining the system under the pressure of an aging population. Adjusting the equivalent payment index can effectively balance the relationship between transitional and basic pensions. Moreover, implementing a policy of delayed retirement could reduce the proportion of transitional pensions within the basic pension, serving as an effective measure to alleviate pressure on the Social Collective Fund.

We analyzed the correlation between the basic pension and its constituent factors, specifically focusing on the standard basic pension, personal account pension, and transitional pension across different post categories (Fig 4). For professional and technical staff, the correlation with the basic pension was ranked as follows: standard basic pension > personal account pension > transitional pension (Fig 4A). For management staff, the ranking was standard basic pension > transitional pension > personal account pension (Fig 4B). For ground-skilled staff, the order shifts to standard basic pension > personal account pension > transitional pension (Fig 4C). Across all three categories, the standard basic pension exhibited the strongest correlation with the basic pension. As both were closely related to an individual's cumulative payment period, our findings suggested that delaying retirement could significantly increase an individual's final basic pension.

### 3.5. Post category and level on the occupational annuity of college retirees during the transition period

We further examined the impact of post grade and post category on retirees' occupational annuities. The findings revealed a positive correlation between post grade and the level of occupational annuity for similar post categories within public institutions during the transition period. For the ground-skilled staff, the average occupational annuity increased from 228.85±172.89 RMB at level IV to 408.17±213.59 RMB at level I, with significant increases observed at levels II and I compared to level IV (Fig 5A). Among the management staff, the average occupational annuity rose from 346.59±118.84

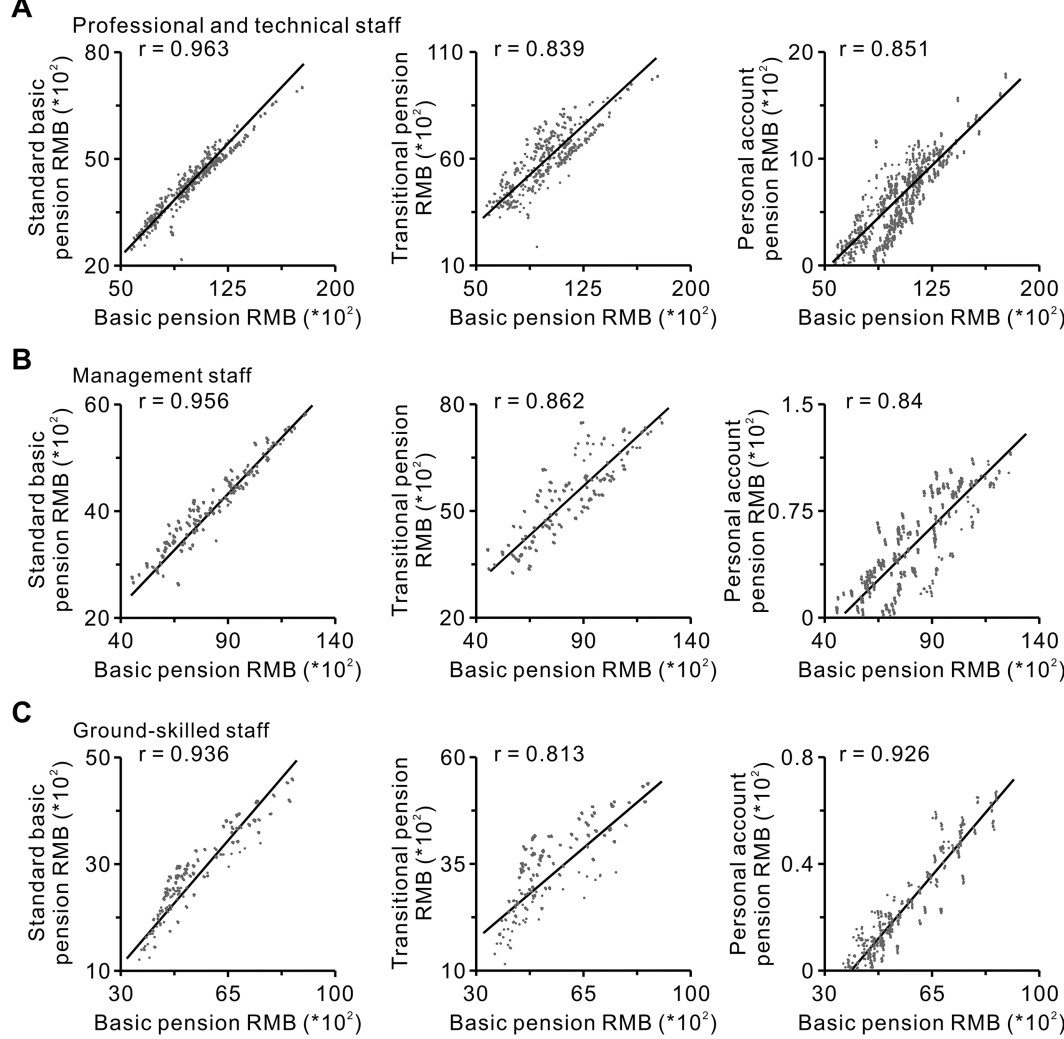

**Fig 4. Correlation between the factors that constitute the basic pension and the basic pension of retirees in different post categories. A:** Correlation between the factors that constitute the basic pension, namely the standard basic pension, transitional pension, and personal account pension, and the basic pension of retirees among professional and technical staff (Pearson correlation test). **B:** Correlation between the factors that constitute the basic pension, namely the standard basic pension, transitional pension, and personal account pension, and the basic pension of retirees in management staff (Pearson correlation test). **C:** Correlation between the factors that constitute the basic pension, namely the standard basic pension, transitional pension, and personal account pension, and the basic pension of retirees among ground-skilled staff (Pearson correlation test).

RMB at level IX to 821.85 ± 365.89 RMB at level IV, with notable increases for managers at levels VII and above compared to level IX (Fig 5A). For the professional and technical staff, the average occupational annuity grew significantly, from 396.13 ± 204.02 RMB at level X to 1427.64 ± 346.58 RMB at level II, with a marked increase observed for staff at levels VI and above compared to level II (Fig 5A).

During the transition period, the average level of occupational annuities varied significantly across the post categories. The professional and technical staff in public institutions received the highest average occupational annuity at 689.76 ± 376.29 RMB, followed by the management staff at 561.64 ± 334.57 RMB, while the ground-skilled staff had the lowest average at 290.37 ± 195.27 RMB (Fig 5B). These findings demonstrated that occupational annuity levels during

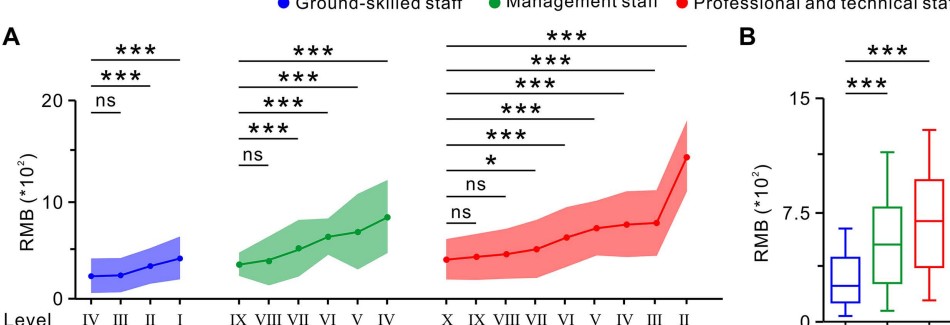

**Fig 5. Impact of post category and grade on retirees' occupational annuities. A:** Variations in occupational annuities among ground-skilled staff, management staff, and professional and technical staff at different levels (one-way ANOVA and post hoc SNK test, respectively). **B:** Differences in occupational annuities between ground-skilled staff, management staff, and professional and technical staff (one-way ANOVA and post hoc SNK test, respectively). Data are presented as mean±SD; ns indicates no significant difference; *$P<0.05$, **$P<0.01$, ***$P<0.001$.

the transition period were influenced by both the post category and post grade. Generally, professional and technical staff received higher occupational annuities than management staff, who received more than ground-skilled staff. Additionally, within each post category, there was a positive correlation between the post level and occupational annuity payments.

## 4. Discussion

Based on the above analysis, the following conclusions can be drawn. During the 10-year transition period following the reform of the basic pension insurance system, pensions for public institution employees who retired before or during the transition period exhibited a steady upward trend. However, due to the introduction of occupational annuities for transition-period retirees, the income gap between the pre-transition and transition retirees did not narrow as anticipated but remained relatively stable (Fig 1). Additionally, the differences in retirement years and accumulative payment periods influenced by gender, post category, and post level significantly affected both basic pension (Fig 2 **and** Table 1) and occupational annuity (Figs 3 and 4). The transitional pension funded by the social pooling system imposed a substantial financial burden, while the accumulative payment periods also played a critical role in determining the basic pension income (Table 1).

The primary findings of this study provide a multidimensional reflection on and extension of theories related to pension system reform and social equity. Firstly, the stabilization of income disparities arising from occupational annuities supports Esping-Andersen's (1990) discussions on path dependence within welfare system theory [4]. This indicates that during periods of institutional transition, the cumulative effects of both legacy systems and new frameworks may reinforce rather than alleviate existing structures of inequality. Secondly, the fiscal pressures faced by the social pooling fund are consistent with Barr and Diamond's (2008) pay-as-you-go pension theory [5], thereby underscoring the challenges posed by uneven regional economic development to the sustainability of pension systems in light of population mobility. Thirdly, regarding policies for postponing retirement, this study's findings lend support to Cahill et al.'s (2015) theoretical framework on flexible retirement [6]. Specifically, gradual adjustments to the retirement age can relieve pressure on pension disbursements while simultaneously promoting a more rational utilization of elderly human resources. Notably, this study's discovery concerning the gender-based pension gap enriches Foster and Walker's (2015) theory on the cumulative process of gender inequality across the life cycle [7]. It reveals how disparities in retirement ages further exacerbate

poverty risks faced by elderly women through mechanisms such as differences in contribution periods and access to occupational annuities.

These findings offer significant theoretical insights for understanding the social implications of China's pension system reform and provide valuable empirical evidence for comparative research on welfare systems in emerging economies. Based on these findings, this study thus proposes several recommendations.

### 4.1. Adjustment of occupational annuity payment rate and level reasonably to ensure fairness within the pension insurance system

Occupational annuity is a vital mechanism for ensuring financial security during retirement, serving as a strategy to motivate employees and attract high-quality talent. Since the reform of the basic pension insurance system on October 1, 2014, public institutions have adopted a pension system that combines basic pension insurance with occupational pension. As occupational annuities are mandatory and form a significant part of the overall pension structure, retirees from public institutions experienced annual income increases during the transition period under current payment rates and payment levels. However, this also widened the income gap between transition-period retirees and those who retired earlier (Fig 1). To address this issue, it is imperative to adjust occupational annuity payment rates and distribution levels to fully utilize their role in ensuring pension security. Such adjustments could help mitigate income disparities between retirees before and during the transition period, thereby fostering greater social equity within China's institutional pension insurance framework.

### 4.2 Improvement of national pooling system for Social Collective Fund and establishment of uniform national standards for payments and pension measurement

Our research revealed that the transitional pension funded by the Social Collective Fund constituted the largest proportion of the monthly payable basic pension (Table 1). Significant disparities in economic development among China's provinces have resulted in variations in employment attractiveness and driving labor mobility across regions [8]. Economically strong provinces have attracted large numbers of young and middle-aged employees, becoming labor-importing regions, whereas weaker provinces face population outflows and serve as labor-exporting regions. Under the current negative population growth trends, labor-exporting provinces are facing escalating challenges in ensuring pension security, as population mobility intensifies the strain on old-age care in these areas. Since the establishment of a central adjustment system for the Social Collective Fund in 2018, total adjustments exceeding 2.5 trillion RMB have alleviated financial pressures on labor-exporting provinces. Although China has achieved national pooling within its pension insurance adjustment system, migration-related complexities still require enhanced coordination. Moving forward, optimizing, and updating institutional arrangements to achieve full national integration within a "unified collection and unified expenditure" framework is an urgent priority in China's basic pension insurance system [9]. Strengthening national pooling and addressing disparities in the Social Collective Fund across provinces are essential strategies for mitigating current and future pension pressure [10].

The transitional pension payment amount funded by the Social Collective Fund is positively correlated with payment and transition coefficients. To ensure consistency in pension benefits across provinces, the government, enterprises, and all sectors of society should collaborate to adjust payment and transition coefficients. Establishing a unified national payment standard and basic pension calculation method tailored to China's national conditions [11] can alleviate the pressure on Social Collective Fund, ensure long-term stability of the pension system, and strengthen its role as a pillar of basic livelihood security. This approach can promote social equity, enhance retirees' quality of life, and contribute to social stability and economic development, ultimately achieving the goal of "providing for the old" while reducing the pension burden on the government [12].

### 4.3. Implementation of a flexible delayed retirement policy and gradual introduction of voluntary retirement at the same age for both men and women

Under United Nations criteria, a country is classified as an aging society when individuals aged 60 and above constitute more than 10% of the total population or when those aged 65 and older exceed 7% [13]. China officially entered an aging society in 2000 according to established criteria. After more than two decades of demographic shifts, the country experienced negative population growth for the first time in 2022 [14,15]. Projections indicate that by 2035, the population aged 60 years and above will exceed 400 million, accounting for over 30% of the total population. This trend highlights the growing severity of population aging in China [16]. Sustained population aging can result in continued negative growth rates, further intensifying demographic aging and exacerbating pension pressures in Chinese society [17,18]. The data analysis in this study revealed that delaying retirement could enhance individual income while alleviating the overall burden on the social pension system. This strategy can promote a mutually beneficial development path for both individuals and society to ensure pension security. Implementing a flexible delayed retirement policy is a viable strategy for addressing pension-related pressure [19,20]. Additionally, postponing retirement can benefit older adults by improving attention span, memory retention, and physical performance, while serving as a preventive measure against common chronic diseases, such as Alzheimer's disease, angina pectoris, and hypertension. This approach can positively contribute to maintaining a higher quality of life in later years [19,21–23].

Data from the China Health and Retirement Longitudinal Study (CHRLS) reveal a significant gender gap in pension benefits, with women receiving less than half of what men receive. Approximately one-third of this disparity is attributable to women's shorter employment duration and lower salaries [24]. Our findings highlighted the substantial differences in basic pension and occupational annuity payments, primarily driven by the different retirement ages of men and women (Figs 2 and 3). Recognizing these disparities, China is implementing a flexible retirement policy that balances individual preferences with efforts to facilitate an orderly transition toward gradually increasing the legal retirement age. The "Global Life Expectancy in 2023" report by the World Health Organization indicates that the average life expectancy in China has risen to 78.21 years, an increase of 9.66 years from 68.55 years in 1990. Specifically, the male life expectancy reached 75.46 years, up by 8.62 years from 66.84 years in 1990, while the female life expectancy averaged 81.16 years, an increase of 10.69 years from 70.47 years in the same period, representing a gender gap of approximately 5.70 years. This trend indicates an increase in China's aging population, with life expectancies expected to continue to rise [25]. Our study randomly selected the retirees from universities, revealing the average retirement ages of 60.5 years for the professional and technical staff, 58.5 years for the management staff, and just under 56.5 years for the support staff. Compared to global averages, China's current retirement age demonstrates the potential for further increases, particularly from the perspective of labor adaptability. Notably, women retire earlier than men while generally enjoying a higher life expectancy. Based on the principle of voluntariness, there is room to extend the retirement age for women to achieve the parity with men. This adjustment would help address gender-based imbalances in post-retirement income [26].

## Supporting information

**S1 Table. Summary of statistics and analysis methods.**
(DOCX)

**S1 File. Data set relevant to this study.**
(XLSX)

## Acknowledgments

We thank all participants and stakeholders for their contributions to this study.

## Author contributions

**Conceptualization:** Xu-Juan Lu, Qian-Xing Zhuang.

**Data curation:** Xu-Juan Lu.

**Formal analysis:** Qian-Xing Zhuang.

**Funding acquisition:** Xu-Juan Lu.

**Investigation:** Xu-Juan Lu.

**Methodology:** Qian-Xing Zhuang.

**Project administration:** Xu-Juan Lu.

**Resources:** Xu-Juan Lu.

**Software:** Qian-Xing Zhuang.

**Supervision:** Xu-Juan Lu, Qian-Xing Zhuang.

**Validation:** Xu-Juan Lu, Qian-Xing Zhuang.

**Visualization:** Qian-Xing Zhuang.

**Writing – original draft:** Xu-Juan Lu.

**Writing – review & editing:** Qian-Xing Zhuang.

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
