## [Decision Letter · Decision Letter 0]

24 Apr 2025

Dear Dr. Zhuang,

The referee believes that there are several important issues that need to be addressed and therefore recommends major revisions. Based on my assessment, I agree with the points raised by the referee. In addition, I believe that the manuscript would benefit from a stronger engagement with the theoretical literature relevant to the topic, particularly in relation to the main findings.

Therefore, based on both my reading of the paper and the referee’s comments, I recommend a weak revise and resubmit decision.

We look forward to receiving your revised manuscript.

Kind regards,

Tamara Fioroni

Academic Editor

PLOS ONE

Journal Requirements:

“This work was supported by the General Projects of Philosophy and Social Sciences in Higher Education Institutions of Jiangsu Province (grants 2022SYB1735)”

**Comments to the Author**

1. Is the manuscript technically sound, and do the data support the conclusions?

Reviewer #1: Partly

2. Has the statistical analysis been performed appropriately and rigorously?

Reviewer #1: I Don't Know

3. Have the authors made all data underlying the findings in their manuscript fully available?

Reviewer #1: No

4. Is the manuscript presented in an intelligible fashion and written in standard English?

Reviewer #1: Yes

Reviewer #1: The article addresses a globally relevant issue: population aging and its implications for pension systems. In particular, the authors focus on the Chinese pension system, analyzing the effects of a reform initiated in 2014 and structured over a ten-year transitional period.

In my opinion, although the topic is relevant and the study presents some interesting aspects, the manuscript requires a major revision before it can be considered for publication.

Among the strengths of the paper, I acknowledge:

• the clear relevance of the subject;

• the size of the dataset, which includes 1,200 retirees from before the transition and 3,270 from during the transition period.

However, several weaknesses compromise the overall quality and clarity of the manuscript. In particular:

1. Formatting issues with mathematical content

The mathematical formulas are presented inline (e.g., page 15), rather than in a proper equation environment. This makes them difficult to read. Furthermore, many symbols are introduced after the formulas, reducing their immediate comprehensibility. The mathematical notation should be cleaned up, with clearly defined variables and standalone, numbered equations where appropriate.

2. Lack of explanation for key techniques

Several methods and indices are used without any introduction or justification. These should be properly explained, especially for non-expert readers.

3. Impression of a rushed submission

The writing gives the impression that the paper was submitted prematurely. A thorough revision is necessary to improve both technical clarity and narrative cohesion. Moreover, clicking on the link to the dataset opens a page in Chinese. It would be better to upload the data to a public repository with an explanation in English, so that access is understandable internationally.

To conclude, I believe the study addresses an important and interesting question, and the data used are valuable. However, the authors should revise the manuscript with greater care and avoid assuming too much background knowledge on the part of the reader.

**Do you want your identity to be public for this peer review?** For information about this choice, including consent withdrawal, please see our Privacy Policy

Reviewer #1: No

---

## [Author Response · Author response to Decision Letter 1]

30 Jun 2025

Response to Reviewer # 1

Many thanks to the reviewer for his/her encouraging comments and valuable suggestions to our manuscript.

The following is our point-by-point reply to the issues raised by the reviewer:

Comments to the Author

1. Is the manuscript technically sound, and do the data support the conclusions?

Reviewer #1: Partly

2. Has the statistical analysis been performed appropriately and rigorously?

Reviewer #1: I Don't Know

3. Have the authors made all data underlying the findings in their manuscript fully available?

Reviewer #1: No

4. Is the manuscript presented in an intelligible fashion and written in standard English?

Reviewer #1: Yes

5. Review Comments to the Author

Reviewer #1: The article addresses a globally relevant issue: population aging and its implications for pension systems. In particular, the authors focus on the Chinese pension system, analyzing the effects of a reform initiated in 2014 and structured over a ten-year transitional period.

In my opinion, although the topic is relevant and the study presents some interesting aspects, the manuscript requires a major revision before it can be considered for publication.

Among the strengths of the paper, I acknowledge:

• the clear relevance of the subject;

• the size of the dataset, which includes 1,200 retirees from before the transition and 3,270 from during the transition period.

However, several weaknesses compromise the overall quality and clarity of the manuscript. In particular:

1. Formatting issues with mathematical content

The mathematical formulas are presented inline (e.g., page 15), rather than in a proper equation environment. This makes them difficult to read. Furthermore, many symbols are introduced after the formulas, reducing their immediate comprehensibility. The mathematical notation should be cleaned up, with clearly defined variables and standalone, numbered equations where appropriate.

Response: According to the Reviewer’s suggestion, we have employed mathematical formulas to illustrate the calculation methods for pre-transition and transition pensions, with each component clearly defined and elucidated in relation to the formulas (Page 6, Lines 138-139, and Pages 7-8 in the revised manuscript). We hope that this approach will facilitate a more accessible reading experience for the audience.

2. Lack of explanation for key techniques

Several methods and indices are used without any introduction or justification. These should be properly explained, especially for non-expert readers.

Response: According to the Reviewer’s suggestion, we have included the statistical methods utilized in the figure legends. Additionally, we have incorporated statistical tables into the manuscript that provide detailed information on the tests employed, sample size (n), and statistical descriptions (S1 Table of the revised manuscript). Furthermore, we have revised what was originally Table 2 to Figure 4 to enhance the clarity and intuitiveness of the results (Figure 4 of the revised manuscript).

3. Impression of a rushed submission

The writing gives the impression that the paper was submitted prematurely. A thorough revision is necessary to improve both technical clarity and narrative cohesion. Moreover, clicking on the link to the dataset opens a page in Chinese. It would be better to upload the data to a public repository with an explanation in English, so that access is understandable internationally.

Response: According to the Reviewer’s suggestion, we have made substantial revisions to the manuscript, which include the incorporation of formulas for pension calculation and a comprehensive definition of each component within the pension framework (Page 6, Lines 138-139, and Pages 7-8 in the revised manuscript). Secondly, we provide a detailed clarification of the statistical methods employed in this study within the methods section and accompanying figure legends. Thirdly, in S1 Table, we present an exhaustive description of the statistics utilized in each figure throughout the manuscript. This includes information on comparison objects, statistical methods applied, sample sizes, and corresponding statistical results (S1 Table of the revised manuscript). Finally, due to the presence of sensitive information related to individual income, we have revised the data availability statement (Page 26, Lines 617-627 in the revised manuscript).

To conclude, I believe the study addresses an important and interesting question, and the data used are valuable. However, the authors should revise the manuscript with greater care and avoid assuming too much background knowledge on the part of the reader.

6. PLOS authors have the option to publish the peer review history of their article (what does this mean?). If published, this will include your full peer review and any attached files.

Do you want your identity to be public for this peer review? For information about this choice, including consent withdrawal, please see our Privacy Policy.

Reviewer #1: No

---

## [Decision Letter · Decision Letter 1]

6 Aug 2025

Dear Dr. Zhuang,

The referee is generally satisfied with the revisions and considers the article almost ready for publication. However, he has suggested that a few minor revisions are still necessary before proceeding.

Based on my own assessment, I agree with the points raised by the referee. In addition, as I mentioned in my previous email, I still do not see a stronger connection with the relevant theoretical literature, particularly in relation to the main findings. I therefore kindly ask you to address this aspect in your revised manuscript.

We look forward to receiving your revised manuscript.

Kind regards,

Tamara Fioroni

Academic Editor

PLOS ONE

Journal Requirements:

Reviewers' comments:

Reviewer's Responses to Questions

**Comments to the Author**

Reviewer #1: All comments have been addressed

2. Is the manuscript technically sound, and do the data support the conclusions?

Reviewer #1: Yes

3. Has the statistical analysis been performed appropriately and rigorously?

Reviewer #1: Yes

4. Have the authors made all data underlying the findings in their manuscript fully available?

Reviewer #1: No

5. Is the manuscript presented in an intelligible fashion and written in standard English?

Reviewer #1: Yes

Reviewer #1: The authors have addressed all of my previous points, and I believe the article is almost ready for publication.

Before proceeding with publication, I suggest making the following final revisions:

- Line 168: Is the % sign necessary?

- Lines 168–169: Move the entire formula for A_PI to line 169.

- Line 178: The subscripts n and n − 1 are too large. If the article was written in Word, I suggest using the equation environment or at least reducing the size of the subscripts and ensuring consistency throughout the article (e.g., on line 180). If the article was written in LaTeX, I suggest using something like $x_n$ and then making adjustments wherever needed.

- Line 202: For consistency with the rest of the article, "presented in S1 Table" should be changed to "presented in (Table S1)", with the parentheses and their content in bold.

**Do you want your identity to be public for this peer review?** For information about this choice, including consent withdrawal, please see our Privacy Policy

Reviewer #1: **Yes: ** Mauro Maria Baldi

---

## [Author Response · Author response to Decision Letter 2]

10 Aug 2025

Response to Editor

Many thanks to the editor for her encouraging comments and valuable suggestions to our manuscript.

The following is our reply to the point raised by the academic editor:

Based on my own assessment, I agree with the points raised by the referee. In addition, as I mentioned in my previous email, I still do not see a stronger connection with the relevant theoretical literature, particularly in relation to the main findings. I therefore kindly ask you to address this aspect in your revised manuscript.

Response: According to the Editor’s suggestion, we have incorporated a discussion regarding the relationship between the primary findings of this manuscript and the relevant theoretical literature in the discussion section (Page 21, Lines 490-513 in the revised manuscript).

Response to Reviewer # 1

Many thanks to the reviewer for his encouraging comments and valuable suggestions to our manuscript.

The following is our reply to the points raised by the reviewer:

Comments to the Author

1. Have the authors made all data underlying the findings in their manuscript fully available?

Reviewer #1: No

Response: In accordance with the submission guidelines, we have included a "Data Availability Statement" at the end of the manuscript, which has been approved by the editorial manager.

2. Before proceeding with publication, I suggest making the following final revisions:

- Line 168: Is the % sign necessary?

Response: Thanks very much for the Reviewer’s suggestion, we have carefully examined the formula and determined that the percentage sign is indeed necessary.

- Lines 168–169: Move the entire formula for A_PI to line 169.

Response: According to the Reviewer’s suggestion, we have relocated the entire formula for A_PI to the subsequent line (Page 8, Line 170 in the revised manuscript).

- Line 178: The subscripts n and n − 1 are too large. If the article was written in Word, I suggest using the equation environment or at least reducing the size of the subscripts and ensuring consistency throughout the article (e.g., on line 180). If the article was written in LaTeX, I suggest using something like $x_n$ and then making adjustments wherever needed.

Response: Thanks very much for the Reviewer’s suggestion, we have reduced the size of the subscripts and ensured consistency throughout the manuscript (Page 8, Lines 179-181 in the revised manuscript).

- Line 202: For consistency with the rest of the article, "presented in S1 Table" should be changed to "presented in (Table S1)", with the parentheses and their content in bold.

Response: According to the Reviewer’s suggestion, we have revised the representation of Table S1 (Page 8, Line 203 in the revised manuscript).

---

## [Decision Letter · Decision Letter 2]

26 Aug 2025

Pension Levels of Chinese Institutions During the Transition Period: A Case Study of Universities

PONE-D-25-04091R2

Dear Dr. Qian-Xing Zhuang,

We’re pleased to inform you that your manuscript has been judged scientifically suitable for publication and will be formally accepted for publication once it meets all outstanding technical requirements.

Kind regards,

Tamara Fioroni

Academic Editor

PLOS ONE

Additional Editor Comments (optional):

Reviewers' comments:

Reviewer's Responses to Questions

**Comments to the Author**

Reviewer #1: All comments have been addressed

2. Is the manuscript technically sound, and do the data support the conclusions?

Reviewer #1: Yes

3. Has the statistical analysis been performed appropriately and rigorously?

Reviewer #1: Yes

4. Have the authors made all data underlying the findings in their manuscript fully available?

Reviewer #1: No

5. Is the manuscript presented in an intelligible fashion and written in standard English?

Reviewer #1: Yes

Reviewer #1: In my opinion, the manuscript has satisfactorily addressed the requested revisions and is now ready for publication.

**Do you want your identity to be public for this peer review?** For information about this choice, including consent withdrawal, please see our Privacy Policy

Reviewer #1: **Yes: ** Mauro Maria Baldi

---

## [Editor Report · Acceptance letter]

PONE-D-25-04091R2

PLOS ONE

Dear Dr. Zhuang,

I'm pleased to inform you that your manuscript has been deemed suitable for publication in PLOS ONE. Congratulations! Your manuscript is now being handed over to our production team.

Kind regards,

on behalf of

Professor Tamara Fioroni

Academic Editor

PLOS ONE